# Evaluation of Nonresponse Bias in a Case–Control Study of Pleural Mesothelioma

**DOI:** 10.3390/ijerph17176146

**Published:** 2020-08-24

**Authors:** Chiara Airoldi, Daniela Ferrante, Dario Mirabelli, Danila Azzolina, Corrado Magnani

**Affiliations:** 1Department of Translational Medicine, Unit of Medical Statistics and Cancer Epidemiology, University of Eastern Piedmont, CPO-Piedmont, 28100 Novara, Italy; daniela.ferrante@med.uniupo.it (D.F.); danila.azzolina@uniupo.it (D.A.); corrado.magnani@uniupo.it (C.M.); 2Department of Medical Sciences, Unit of Cancer Epidemiology, CPO Piemonte and University of Turin, 10126 Turin, Italy; dario.mirabelli@gmail.com; 3Interdepartmental Centre G. Scansetti for Studies on Asbestos and other Toxic Particulates, University of Turin, 10125 Turin, Italy

**Keywords:** nonrespondents, record linkage, agreement, asbestos

## Abstract

Nonparticipation limits the power of epidemiological studies, and can cause bias. In a case–control study on pleural malignant mesothelioma (MM), we found low participation in interviews (63%) among controls. Our goal was to characterize nonresponder controls and assess nonresponse bias in our study. We selected all nonresponder controls (204) and a random sample of responder controls (174). Data were obtained linking hospital admissions and town registrars, and concordance between sources was assessed. Nonresponse bias was evaluated using a logistic regression model applying the inverse probability weighting approach. The odds ratio (OR) for the status of the respondents was 0.61 (95% confidence interval (CI): 0.33–1.16) for controls aged 61–70, 0.37 (CI: 0.20–0.66) for those aged 71–80, and 0.40 (CI: 0.20–0.80) for those aged above 80 (reference group: ≤60 years). Controls with low education level had lower OR (0.47; CI: 0.26–0.84). After adjustment, the ORs for MM by categories of cumulative exposure to asbestos were similar to the unadjusted results, ranging from 4.6 (CI: 1.8–11.7) for cumulative exposures between 0.1 and 1 f/mL-y to 57.5 (CI: 20.2–163.9) above 10 f/mL-y. Responder controls were younger and had higher education level. Nevertheless, there was little evidence of bias from nonresponse in the risk estimates of MM.

## 1. Introduction

The population of Casale Monferrato, about 40,000 inhabitants, has an extremely high incidence of malignant pleural mesothelioma (MPM) as a consequence of the activity of a large asbestos cement plant in 1907–1986 [1,2,3,4]. MPM incidence rates (microscopically confirmed cases per 100,000 person-years) in 2011–2015 reached 71.9 in men and 40.2 in women. MPM rates are also very high in the corresponding Health District of Casale Monferrato (48 municipalities for a total population of about 100,000), where rates average 48.6 in men and 25.5 in women, compared with corresponding regional rates of 5.5 and 1.9, respectively [5,6].

Ferrante et al. [7] published the results of a population-based case–control study on MPM carried out in 2000–2006 in the Health District of Casale Monferrato. This study evaluated asbestos exposure from occupational, domestic, and residential sources. Its results showed a clear dose–response relationship with cumulative asbestos exposure: the odds ratio (OR) of MPM increased from 4.4 (95% CI 1.7–11.3) in the lowest cumulative exposure category (0.1–1 fiber/milliliter-year (f/mL-y)) to 62.1 (95% CI 22.2–173.2) in the highest cumulative exposure category (>10 f/mL-y).

In case–control studies, when data are collected from participants (e.g., using questionnaires) [8], it is not uncommon that a relevant proportion of eligible subjects does not agree to participate. Participation of cases and controls may also differ. In the study of Ferrante et al. [7], participation was close to 90% for cases and 63% for controls.

Nonparticipation affects the power of the study and, more importantly, can cause selection bias if it is differential in respect to exposure. Nonparticipation in methodological studies has been associated with low education level, old age, and lack of social and familial relationships [9,10].

In the framework of Ferrante et al.’s study, particular attention should be paid to age and education level. Italians born in the 1930s to 1950s have the highest age-specific mesothelioma mortality rates [11] and are expected to have experienced asbestos exposure at work more frequently due to the rise and fall of asbestos consumption in Italy [12]. Selective participation of controls by age attained during the study recruitment period (2000–2006) might have thus excluded from interview and analyses the controls most likely to have had the exposure of interest. The same line of reasoning applies to education level, as a proxy of socioeconomic status: as late as 1990–1993, approximately 24% of laborers in Italy had some occupational exposure to asbestos, according to the CAREX survey [13].

The aim of the present investigation was to (i) characterize responder and nonresponder controls according to social, demographic, and health-related variables obtained from administrative databases; (ii) assess the association of such variables with the response status; and (iii) provide new, adjusted estimates of the odds ratio of MPM by cumulative exposure to asbestos in the 2000–2006 case–control study.

## 2. Materials and Methods 

### 2.1. Study Design

A population-based case–control study was conducted in the Health District of Casale Monferrato. The cases were subjects with an incident diagnosis of MPM between 1 January 2001 and 30 June 2006. The controls were a random sample of the residents at the time of case diagnosis, matched by date of birth (±18 months) and sex. The sample included 223 MPM cases and 552 controls. Further details have been given by Ferrante et al. [7].

A detailed face-to-face interview based on a standard questionnaire provided information on the occupational history of the study subjects and their family members, as well as on their residential history, house characteristics, and surrounding environment. Out of the 223 cases and 552 controls, the participants (i.e., those who provided the interview, either in person or by a surrogate respondent from close family members) numbered 200 (89.69%) and 348 (63.04%), respectively.

To investigate the determinants and effects of nonparticipation among the controls (as it was negligible among the cases), all the 204 nonrespondents and a stratified random sample of 50% of the respondents, for a total of 378 subjects, were included. Figure 1 presents the procedures and the number of subjects in the different categories.

### 2.2. Data Sources

Two sets of data were used to provide sociodemographic and health-related information on the subjects included in the study: hospital admission records (HARs) and the town registrar’s records (TRs). The two datasets are routinely and independently collected for administrative purposes, and data for the respondents and nonrespondents were recorded. For the respondents only, relevant information from this administrative database was compared with that from the questionnaire.

We considered the HARs from Piedmont hospitals in 1996–2006 (i.e., the study period and the 5 previous years) because HARs were not available in electronic form before 1996. Information included demographic, social (marital status, education, occupational title), and clinical data (coded diagnosis and length of stay). Diagnoses of interest in the clinical data were those pertaining to the following categories: malignancies (ICD-9-CM codes 140.0–208.9), cardiovascular diseases (ICD-9-CM codes 390.0–459.9), respiratory diseases (ICD-9-CM codes 460.0–519.9), digestive diseases (ICD-9-CM codes 520.0–579.9), and accidents and violence (ICD-9-CM codes 800.0–999.9). 

Marital status, education, and occupational status were also retrieved from the TR files. For each subject included in the study, information was obtained by the municipality where the subject was a resident at the time of the study.

Marital status, education, and occupational status were also available from the questionnaire.

All subjects gave their informed consent for inclusion before they participated in the study, and the study was conducted in accordance with the Declaration of Helsinki. The study protocol was not submitted to an ethical committee because approval for observational studies was not needed in the first years of the 2000s in Italy.

### 2.3. Linkage Procedure

For all sources, the name, surname, date of birth, and fiscal identification code (a code used to identify each resident in Italy) were available. 

The first phase was the data cleaning process, including the correction of invalid fiscal codes. HAR files were then linked to the list of subjects in the study using a deterministic and a probabilistic record linkage. The deterministic linkage was based on the perfect agreement between the name, surname, and date of birth or fiscal identification code, while the probabilistic linkage considered the first characters of the surname, name, and birth year. To maximize the true positive, the results of the probabilistic linkage were manually controlled.

### 2.4. Statistical Analysis

For statistical analyses, the variables of interest were grouped in classes as follows: education in two classes (high: university and high school, low: primary and middle school), civil status in three classes (married, widow/widower and divorced, unmarried), age in four classes (≤60, 61–70, 71–80, >80 years), and occupational status in three classes (employed, unemployed, and retired). The number of hospital admissions was classified in three classes (0, 1, >1). For each variable, we reported absolute and relative frequencies separately for the respondents and nonrespondents. Chi-square tests were computed to test for the difference between the respondents and nonrespondents.

For the interviewed subjects, information on marital status, education, and occupational status from HARs and TRs were compared with that collected from the interview, and Cohen’s kappa was computed. The main source to abstract demographic and social data was chosen based on the best value of this index.

Univariable models were used to analyze the likelihood of participation in the study, considering each variable separately. Then, multivariable logistic models, adjusted for age and sex, were fitted. 

A forward selection procedure based on the likelihood ratio test was implemented to select the covariates significantly associated with the response status. The final model was used to estimate the predicted probability of participation for the responder and nonresponder controls. To investigate the hypothesis that propensity to respond might behave differently over the various layers of gender and age class, a logistic model including the interaction term was performed. The subjects for whom data were not obtained from the town registrar’s records or HARs were not included in these models.

To assess the potential bias from nonresponse, we applied the inverse probability weighting approach to the logistic models fitted by Ferrante et al. [7]. Briefly, we calculated the inverse of the predicted probability previously estimated for the responder controls included in this study and with complete information, and we used these values as weight. A weight equal to 1 was assigned to the other responder controls (subjects not included in the random sample or included but with missing data) and the responder cases. 

For each analysis, the odds ratios (ORs) and their 95% confidence intervals (95% CIs) were computed. Statistical significance was set at 5%. Analyses were conducted using SAS 8 (SAS Institute Inc., Cary, NC, USA) and Stata 12 (StataCorp LLC, College Station, TX, USA) [14].

## 3. Results

The 378 subjects we studied (all 204 nonresponder controls and a stratified random sample of 50% of the responder controls) were mainly men (*n* = 226; 59.8%) and adults/elderlies (mean age 66.8 years; standard deviation 12.2). This information, available at the beginning of the study, reflects the epidemiological distribution of malignant mesothelioma.

The HAR database from 1996 to 2006 consisted of 9,919,921 initial records, corresponding to 9,908,188 hospital admissions. The deterministic linkage procedure with the name, surname, and date of birth as a key captured 903 records, while 24 additional records were linked using the fiscal code. The probabilistic procedure identified 1249 records, of which only 80 were considered valid and were added to the record linkage results. Eventually, one or more hospital admission records were found for 260 subjects.

The distribution of education, employment, and civil status obtained from the different sources is shown in the Appendix A. Cohen’s kappa index was calculated to assess the concordance between the available sources and the questionnaire. Considering the town registrar’s records, the highest value of the kappa index was achieved for education (0.83), that relating to marital status (0.76) could be considered good, while only a moderate agreement was observed for employment (0.50). Using HARs, good agreement emerged for marital status (0.72) and education (0.71), while agreement was lower for employment (0.50). Based on these results, the first source of information was the TRs, followed by the HARs. More details are shown in Appendix A.

Sociodemographic variables for the responder and nonresponder controls are shown in Table 1. Significant differences were observed for age, occupation, and education. Most nonrespondents were 70–80 years old (*n* = 70, 34.3%) or older (*n* = 39, 19.1%), while the respondents were mainly ≤60 years old (*n* = 76, 43.7%) (*p*-value = 0.0003). Employed subjects were 30.5% (*n* = 53) of the respondents, compared to 18.1% (37) of the nonrespondents (*p* = 0.006). Both respondents and nonrespondents had a high proportion of subjects with a primary/middle school certificate: 60.3% and 72.0%, respectively (*p*-value = 0.001). No statistically significant differences were found for gender (*p*-value = 0.209) or civil status (*p*-value = 0.374). 

Table 2 reports the clinical variables obtained from the HARs. Two hundred and sixty subjects had at least one hospital admission between 1996 and 2006: 66 (17.5%) had only one and 194 (51.3%) had more than one. Hospital admissions for cardiovascular diseases were significantly more frequent in nonrespondents (*p* = 0.015), and they generally had more comorbidities compared with the respondents. No other statistically significant differences were found.

Results of the multivariate logistic model considering the study participation as outcome are reported in Table 3. The final model included education, gender, and age as covariates. The subjects with lower education level showed an OR equal to 0.47 (95% CI: 0.26–0.84). Age was inversely associated with the response status: the OR was equal to 0.61 (95% CI: 0.33–1.16) for those aged 61–70 years, 0.37 (95% CI: 0.20–0.66) for those aged 71–80, and 0.40 (95% CI: 0.20–0.80) for those above 80, considering people up to the age of 60 as reference. Gender was not associated with the response status. The interaction term in the model was not statistically significant (*p* = 0.075).

The results of the inverse probability weight approach applied to the logistic models fitted by Ferrante are shown in Table 4. The original analyses showed a constant trend of increasing OR with increasing cumulative exposure index; the OR was already significantly increased in the lowest category of cumulative exposure, corresponding to the “up to 1 f/mL-years” class. After inverse probability weight adjustment, the same trend was replicated: the ORs were 4.6 (95% CI 1.8–11.7), 15.5 (95% CI 6.3–38.2), and 57.5 (95% CI 20.2–163.9) for ≥0.1–<1, ≥1–<10, and ≥10 fiber/mL-years, respectively (ref: <0.1).

## 4. Discussion

Although it is often overlooked, nonresponse is an important problem in epidemiological research with a future impact, as declining response rates since the early 2000s have been reported [15,16]. When high percentages of nonrespondents are observed, risk estimates could be biased, and incorrect interpretation can follow. To evaluate the impact of nonresponse bias, a description of respondents and nonrespondents and an adjustment of results are recommended.

In our work, first, we collected information from administrative data on controls, regardless of their participation in the study, and second, considering the difference among them, we weighted the models in the analyses of the association of asbestos and mesothelioma for the inverse probability of being a responder.

This study highlights the association of participation in the study with demographic and personal factors. Advanced age is a risk factor for being a nonrespondent. This result is consistent with the findings of Hill et al. [17], who observed that older people are less inclined to participate in studies and that increasing age is associated with a decrease in response rates. Other studies found a similar association [10,18]. The influence of gender appears less clearly defined in literature. Cochrane reported a low response rate among women in some population studies, while Gordon and Kannel reported that men were less likely to participate [19,20]. Slattery et al. found no gender difference in nonresponse [21]. The hypothesis of interaction between age and gender on the propensity to respond was not supported based on our data. 

The nonrespondents in our study had lower education level and were more likely to be retired. It has been repeatedly reported that individuals with low education level are less inclined to participate [10] and that higher socioeconomic status is a predictor of willingness to respond [18,22,23]. In assessing nonresponse, the marital status of the subjects was considered. In this study, participation OR showed a lower propensity to participate for widowers/separated and unmarried subjects with respect to married subjects. Our results agree with those of Alderson, who argued for lower response rates if the subjects live alone, have no children, or are widowed or divorced [24].

The assessment of the health status of an individual is perhaps the most difficult aspect to consider because there is no univocal way of describing it and data sources providing health information are limited. In previous reports, the focus was more often on cardiovascular diseases, respiratory diseases, or specific pathologies of interest in a particular study, and the evidence of their effect is variable and partially contradictory. Some authors [10,25] observed an association between the presence of cardiac pathologies and low participation, while opposite results were found in the study of Maclure and Hankison [23], where heart disease was associated with control participation. In a study about diabetic subjects [10], the presence of retinopathy and nephropathy were not associated with respondence. In a prevalence study on respiratory health conducted in Norway by Abrahamsen et al., no significant associations were observed between respondence and the presence of asthma and other respiratory diseases [25]. In this study, hospital admission because of cardiovascular diseases was the only health condition that, in the univariate analysis, was significantly more frequent among nonresponder controls. This difference is most likely attributable to the older age of nonrespondents. When the cardiovascular disease variable was included in a model adjusted for gender and age, this association disappeared.

Our study conclusions are consistent with those of an Italian case–control study on lung cancer: older age, lower education level, and being unmarried (widowed or single) are significant risk factors for being nonrespondents [9].

These differences in participation could cause a distortion in the estimates if the age and education level variables were associated with the exposure. The effects of nonresponse were thus accounted for using inverse probability weighting. The obtained OR estimates fitting the model closely matched the results obtained by Ferrante et al. [7], indicating that nonresponse bias does not change the direction of the association between asbestos and incidence of malignant mesothelioma (MM). The weighting appeared to improve the accuracy of the estimates—confidence intervals’ width was generally reduced. 

The most critical aspects of this study are the choice of variables and the methods of data collection. The residences of the subjects, their economic conditions, or other lifestyle aspects could further differentiate respondents and nonrespondents and potentially be associated with asbestos exposure, but no information was available to us. The sources of data equally available for respondents and nonrespondents had limitations because they were conceived for different purposes from medical research. HARs are originally collected for the governance of hospitals and the compensation of diagnosis-related groups, and provide the main disease diagnosis plus up to five accessory diagnoses coded according to the Ninth Revision of the International Classification of Diseases. TRs are used to enumerate citizens at the town level, and associate only relatively crude information on their civil status, education level, and occupation. Furthermore, we searched TRs about 10 years after the study had been completed (as well as personal interviews). We assume that differences in education level are negligible considering the age distribution of the study subjects, but some differences of occupation and civil status may exist between the value when we extracted information from TRs and that at study eligibility. Particularly, a larger number of individuals were classified as retired and widowed in TRs data compared with interview data. However, we assume that time acted similarly between respondents and nonrespondents. 

Multivariate analyses were carried out, including age, gender, and education conditions in the inverse probability weight, in order to adjust for potential confounding and related non-collapsibility bias; the resulting model was redundant, as gender did not show a statistically significant association with study participation. 

The choice to consider only a sample of responder controls (50%) was due to the available resources, both in the research team and in the municipal offices interested in the investigation. Particularly, about 60 different municipal offices were contacted for the research, and for each town, information on many subjects was required; so, increasing the study size was expected to slow down or even reduce compliance and, thus, study feasibility. Moreover, the random sampling and the large number of subjects provided unbiased information and adequate power of the analysis.

## 5. Conclusions

Our new analyses of mesothelioma risk following asbestos exposure, adjusted for nonparticipation among controls, provided results strictly consistent with those already published by Ferrante et al. [7], and showed that nonparticipation did not cause major bias in this study. 

## Figures and Tables

**Figure 1 ijerph-17-06146-f001:**
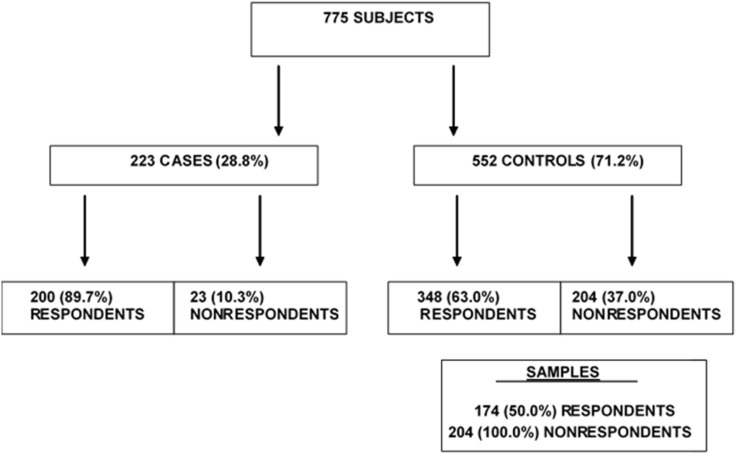
Flow chart of the selection process of the subjects in the case–control study and the responder and nonresponder controls included in the study.

**Table 1 ijerph-17-06146-t001:** Descriptive information (*n* and %) for the respondent and nonrespondent controls: sociodemographic variables.

Variables	Respondents		Nonrespondents		Total	
	*n*	%	*n*	%	*n*	%
**Gender**						
Female	64	36.8	88	43.1	152	40.2
Male	110	63.2	116	56.9	226	59.8
**Age**						
≤60	76	43.7	53	26.0	129	34.1
60–70	41	23.6	42	20.6	83	22.0
70–80	34	19.5	70	34.3	104	27.5
>80	23	13.2	39	19.1	62	16.4
**Occupation**						
Employed	53	30.5	37	18.1	90	23.8
Unemployed	19	10.9	37	18.1	56	14.8
Retired	91	52.3	121	59.3	212	56.1
Missing	11	6.3	9	4.4	20	5.3
**Civil status**						
Married/cohabitant	111	63.8	117	57.3	228	60.3
Widowed/Separated	44	25.3	62	30.4	106	28.0
Unmarried	15	8.6	22	10.8	37	9.8
Missing	4	2.3	3	1.5	7	1.9
**Education level**						
Primary/middle school	105	60.3	147	72.0	252	66.7
Secondary school/university	43	24.7	23	11.3	66	17.5
Missing	26	14.9	34	16.7	60	15.9
**Total**	**174**	**100.0**	**204**	**100.0**	**378**	**100.0**

**Table 2 ijerph-17-06146-t002:** Descriptive information (*n* and %) for the respondent and nonrespondent controls: clinical variables from the hospital admission records. The analyses were restricted to the controls with at least one hospital record: 120 (46.2%) and 140 (53.8%) for the respondents and nonrespondents, respectively.

Variables	Respondents		Nonrespondents		Total	
	*n*	%	*n*	%	*n*	%
**Hospital records**						
0	54	31.0	64	31.4	118	31.2
1	33	19.0	33	16.2	66	17.5
>1	87	50.0	107	52.4	194	51.3
**Total**	174	100.0	204	100.0	378	100.0
**Malignant neoplasm**						
No	100	83.3	110	78.6	210	80.8
Yes	20	16.7	30	21.4	50	19.2
**Cardiovascular diseases**						
No	66	55.0	52	37.1	118	45.4
Yes	54	45.0	88	62.9	142	54.6
**Respiratory diseases**						
No	95	79.2	109	77.9	204	78.5
Yes	25	20.8	31	22.1	56	21.5
**Digestive diseases**						
No	79	65.8	86	61.4	165	63.5
Yes	41	34.2	54	38.6	95	36.5
**Accidents and violence**						
No	106	88.3	122	87.1	228	87.7
Yes	14	11.7	18	12.9	32	12.3
**Total**	**120**	**100.0**	**140**	**100.0**	**260**	**100.0**

**Table 3 ijerph-17-06146-t003:** Logistic regression model (outcome: probability of response). The odds ratios (ORs) and the 95% confidence intervals (CIs 95%) were obtained using a multivariable model where gender and age were included regardless of statistical significance.

Variables	Levels	Total	Multivariable Model
		*n*	OR (CI 95%)
Gender	Female	152	1 (ref)
	Male	226	0.87 (0.54–1.39)
Age	≤60	129	1 (ref)
	61–70	83	0.61 (0.33–1.16)
	71–80	104	0.37 (0.20–0.66)
	>80	62	0.40 (0.20–0.80)
Education level	Primary/middle	252	0.47 (0.26–0.84)
	High/university	66	1 (ref)

**Table 4 ijerph-17-06146-t004:** Risk of malignant pleural mesothelioma in relation to the asbestos cumulative exposure index. All models were adjusted for age, gender, and type of interview. We reported the original odds ratios, 95% confidence intervals (95% CIs), and inverse probability weight-adjusted ORs (ORadj).

Cumulative Exposure to Asbestos Index (in Fiber/mL-Years)	Cases *n* (%)	Controls *n* (%)	OR (95% CI)	ORadj (95% CI)
Background level (<0.1)	8 (4.0)	106 (30.5)	1 (ref)	1 (ref)
≥0.1–<1	26 (13.0)	108 (31.0)	4.4 (1.7–11.3)	4.6 (1.8–11.7)
≥1–<10	113 (56.5)	115 (33.0)	17.5 (7.3–41.8)	15.5 (6.3–38.2)
≥10 (range: 10–4128)	53 (26.5)	19 (5.5)	62.1 (22.2–173.2)	57.5 (20.2–163.9)
Total	200 (100.0)	348 (100.0)

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
