# Peer review of "Evaluation of Nonresponse Bias in a Case–Control Study of Pleural Mesothelioma"

_ijerph, 2020, doi:10.3390/ijerph17176146_

Round 1
Reviewer 1 Report
The manuscript “Evaluation of Non-Response Bias in a Case-Control Study of Pleural Mesothelioma” revisits a previous paper published in 2016 written by the same group with the objective of evaluating the non-response bias due to partial participation of controls (63%).
Data sources available for searching variables possibly related to study bias were hospital admission records (HAR) and population prefecture records, both administrative databases. The HAR records were restricted for the available period (1996 onwards). Linkage between databases was basically deterministic. The authors were careful in choosing the best source to retrieve sociodemographic data.
Statistical methods are well described and results are clearly presented. The discussion is clear and the conclusion is supported by the study findings.
In the last paragraph of the discussion it is stated that the choice of analyzing 50% of the respondents was due to the available resources. My understanding is that the data was readily available for the whole group. I guess that including all respondents would not change the results because the authors stratified the respondents sample. But it would be a minor extra-work of checking the linkage procedure if the whole group of respondents were included.
Author Response
The manuscript “Evaluation of Non-Response Bias in a Case-Control Study of Pleural Mesothelioma” revisits a previous paper published in 2016 written by the same group with the objective of evaluating the non-response bias due to partial participation of controls (63%).
Data sources available for searching variables possibly related to study bias were hospital admission records (HAR) and population prefecture records, both administrative databases. The HAR records were restricted for the available period (1996 onwards). Linkage between databases was basically deterministic. The authors were careful in choosing the best source to retrieve sociodemographic data.
Statistical methods are well described and results are clearly presented. The discussion is clear and the conclusion is supported by the study findings.
In the last paragraph of the discussion it is stated that the choice of analyzing 50% of the respondents was due to the available resources. My understanding is that the data was readily available for the whole group. I guess that including all respondents would not change the results because the authors stratified the respondents sample. But it would be a minor extra-work of checking the linkage procedure if the whole group of respondents were included.
We thank the Reviewer for highlighting the many strengths of our proposal, including the statistical methods and the data source used. However, as remarked by the Reviewer, we decided to analyse only 50% of the responders for two reasons. First, the Town Registrar’s Offices data was collected and archived by different Municipalities, based on the town of residence of the study subjects. The process leading to information collection is not automatic and requires interaction between the researchers and the municipal offices staff. About 60 municipal offices were contacted for the research, and for each city many subject’s information was required. For example, more than 100 patients came from the city of Casale Monferrato. We had to limit our requests according to the limits set by the interested municipalities. Second, we completely agree with the Reviewer’s hypothesis that including all responders would not have changed the results as the stratified sampling was randomized.
We added a sentence (lines 274-280) to address these points.
Does the introduction provide sufficient background and include all relevant references? Can be improved
We added further considerations (lines 50-58) and two new references (current refs 11, 12 and 13).
Reviewer 2 Report
I was pleased to be asked to see this paper. I have looked after patients with mesothelioma all my professional career as a surgeon. I have no personal expertise in epidemiology but I have worked and published with epidemiologists and others who work with big data and cancer registries.
The problem of unexpected or hidden biases is very great. They can undermine small studies such as clinical reports but as the authors point out big data sets are far from immune. I learned a lot from it.
As far as I can judge, this is a very well-considered paper with a well-researched introduction and discussion, and well documented methods and results. There is scientific rigour at a much higher level than I see amongst my surgeon colleagues – but no advantage in mentioning that to them – it isn’t what they do. Sparing these patients unavailing surgery was my contribution and doing what I could to palliate where it was possible.
Author Response
I was pleased to be asked to see this paper. I have looked after patients with mesothelioma all my professional career as a surgeon. I have no personal expertise in epidemiology but I have worked and published with epidemiologists and others who work with big data and cancer registries.
The problem of unexpected or hidden biases is very great. They can undermine small studies such as clinical reports but as the authors point out big data sets are far from immune. I learned a lot from it.
As far as I can judge, this is a very well-considered paper with a well-researched introduction and discussion, and well documented methods and results. There is scientific rigour at a much higher level than I see amongst my surgeon colleagues – but no advantage in mentioning that to them – it isn’t what they do. Sparing these patients unavailing surgery was my contribution and doing what I could to palliate where it was possible.
We sincerely appreciate the Reviewer’s comments and we are joyful to actively contribute to the scientific research about asbestos-related diseases.